# Teaching RAG to Play Fair: Assessing and Mitigating Encoder-Only PLM Algorithmic Bias

## 1    Introduction

Large language models (LLMs) have demonstrated remarkable capabilities in generating high-quality texts for a wide range of purposes. However, they often hallucinate facts and struggle to produce responses grounded in reliable external knowledge. Retrieval-Augmented Generation (RAG) have emerged as a promising solution to assist the LLMs. Augmenting LLMs with access to large corpora of documents at inference time, RAG can reduce the hallucinations while enhancing factual accuracy of the generated content (Hu et al., 2024a). Across diverse tasks, RAG has been shown to enable the use of smaller LLMs while reducing false claims (Jin et al., 2024).

Central to this retrieval process is the encoder-only pre-trained language model (PLM), which computes representations of both user queries and candidate documents in semantic vectors. These representations determine the most relevant documents from the document corpus and passed to the LLM as evidence for accurate generation (Li et al., 2022).

While RAG systems have shown great potential for fulfilling their goal, their fairness remains an emerging concern. In particular, despite its essential role, the fairness concerns for the encoder component has received little attention in fairness research. Specifically, most existing fairness research in RAG focuses on biases in the generated outputs or the content of retrieved documents, overlooking how the encoder itself can shape retrieval outcomes through its internal representations. The biased representation in encoder is particularly concerning as it could more easily propagate the bias to the generation stage (Hu et al., 2024b). This oversight is critical: if an encoder embeds gender, racial, or political bias into its similarity space, then even from an unbiased corpus, the documents retrieved can be skewed in ways that disproportionately misrepresent or under-represent certain groups (Cao and Zhang, 2025). Additionally, the lack of modular debiasing tools for encoder-only models makes it difficult to isolate and mitigate unfair behavior systematically caused by specific components, making the existing problem especially difficult to solve (Hu et al., 2024b).

This potentially leads to what fairness scholars term **disparate impact**: users from different demographic groups may receive systematically different answers, not due to explicit discrimination in corpus or the generative model, but because of biased structuring in the semantic space during retrieval (Mack, 2023).

Furthermore, such representational bias may not manifest as explicit toxicity or offensive content but can subtly distort semantic associations in the embedding space. For example, the encoder can produce vector representations that consistently place tokens related to women closer to concepts like "home" or "family", while associating certain racial or ethnic terms more closely with "crime" or "violence". These biased patterns influence how the system ranks and selects documents, resulting in **systematic disparities** in the information users receive. Notably, such bias can persist even when the training data is balanced, because the model tends to reinforce subtle patterns and associations already present in the data (Bolukbasi et al., 2016).

## 2    Research Objectives

This study investigates representation-level bias and debiasing techniques in encoder-only pre-trained language models (PLMs) within RAG systems. The primary research question and other secondary research questions are formulated as follows:

**Primary Research Question:**
*How can bias in encoder-only pre-trained language models used in Retrieval-Augmented Generation (RAG)(or referred as retriever in the context of RAG) be identified and mitigated at the representation level, without relying on downstream task outputs?*

Importantly, out work is distinguished from other studies that investigate bias in the generation component of RAG models: our focus is exclusively on the retriever, examining bias in the encoder representations rather than in the text produced by the generator.

**Sub-questions:**

- **RQ1:** Where and how is social bias encoded in the representations of encoder-only PLMs used in RAG systems?
- **RQ2:** Do current representation-level fairness metrics adequately reflect algorithmic bias in encoder models?
- **RQ3:** How far can existing lightweight adaptation methods effectively debias encoder representations without degrading retrieval quality?
- **RQ4:** How can a systematic debiasing framework built on top of the existing methods be designed and applied to encoder-only PLMs in RAG systems to mitigate representational bias while preserving retrieval performance?

## 3 METHODOLOGY

### 3.1 EXPERIMENTAL SETUP

#### 3.1.1 MODELS SELECTION

To control confounds from varying backbones, we standardize on a single encoder for all research questions: **all-MiniLM-L6-v2** (Sentence-Transformers).

1. **all-MiniLM-L6-v2**. A compact, widely adopted encoder that offers a strong quality–efficiency trade-off on the **MTEB** leaderboard among lightweight models (Muennighoff et al., 2023). It produces *384-dimensional* sentence embeddings and runs efficiently on CPU and commodity GPUs, enabling large-scale experiments and multiple ablations under fixed compute and memory budgets. We select this model because it is a well-established baseline in production-grade RAG systems, and competitive on retrieval and semantic similarity tasks relative to its footprint.

#### 3.1.2 DATASET SELECTIONS

For RQ1 and RQ2 we use the **Bias Benchmark for Question Answering (BBQ)** dataset (Parrish et al., 2022), which contains about 58k multiple-choice questions covering nine protected categories (e.g., gender identity, race/ethnicity, socioeconomic status). BBQ is especially well-suited for evaluating representational and output-level bias, allowing us to link encoder representations with disparities in downstream QA. Details of how we adapt BBQ for intrinsic metrics are provided in Appendix 6.2.

For RQ3 and RQ4 we use a sentence-level Wikipedia dump (2023-03-01 snapshot) (Emmer, 2023) as the retrieval corpus, due to its tractable size and broad coverage. We construct training and evaluation datasets by pairing demographic contrasts from BBQ with topical queries and retrieving documents from this corpus. The data generation pipeline, including group pairing, prompting, and rephrasing into neutral and biased variants, is described in Appendix 6.2.

### 3.2 BIAS DIAGNOSIS AND LOCALIZATION (RQ1)

For the bias diagnosis of the Encoder, we will utilize the intrinsic metrics introduced in the related work section, which are the distance-based metrics and probing based metrics. The dataset used for this section would be the Gender identity subset of the BBQ dataset.

### 3.2.1 DISTANCE-BASED METRICS

We utilize the **Sentence Embedding Association Test (SEAT)** (May et al., 2019), a distance-based fairness diagnostic designed to quantify *associative bias* in sentence embeddings.

Given sentence embeddings $X = \{x_1, \ldots, x_m\}$ and $Y = \{y_1, \ldots, y_n\}$ as target sets, and attribute embeddings $A = \{a_1, \ldots, a_k\}$ and $B = \{b_1, \ldots, b_k\}$, the association of a sentence embedding $w$ is defined as:

$$s(w, A, B) = \frac{1}{|A|} \sum_{a \in A} \cos(w, a) - \frac{1}{|B|} \sum_{b \in B} \cos(w, b) \tag{1}$$

And the SEAT effect size, which serves as our primary measure of representational bias from (May et al., 2019), is:

$$\text{EffectSize} = \frac{\mathbb{E}_{x \sim X}[s(x,A,B)] - \mathbb{E}_{y \sim Y}[s(y,A,B)]}{\sqrt{\text{Var}_{w \sim (X \cup Y)}[s(w,A,B)]}} \tag{2}$$

We define the **fairness score** from SEAT as:

$$\text{FairnessScore}_{\text{SEAT}} = 1 - \frac{|\text{EffectSize}|}{E_{\max}} \tag{3}$$

where $E_{\max}$ is a normalization constant (we set this to 2.0) to bound the score in $[0, 1]$. A higher score indicates better fairness, with 1.0 denoting perfectly neutral representations. By observing the performance of the models of this metrics on the selected Gender identity dataset, we can then determine the extent of bias on each of the models

### 3.2.2 PROBING CLASSIFIERS

Moreover, we need to answer where the bias is in the model by localizing which layer is the protected-attribute information encoded in the model. To this end, we employ **probing classifiers** (Gupta et al., 2024).

To measure how stereotypical bias is encoded in different layers of a language model, we adopt a regression-based probing approach that predicts a model-internal bias score constructed from contextualized embeddings.

Given a dataset of stereotype-sensitive questions $q$ and contexts $c$, we define the following model inputs: the concatenated input $x_{cq} = q\|c$, the stereotyped instantiation $x_s = q\|g_s$, and the non-stereotyped instantiation $x_n = q\|g_n$, where $g_s$ and $g_n$ denote the stereotyped and non-stereotyped group labels, respectively.

Using a frozen encoder model, we obtain mean-pooled embeddings from each layer $\ell$ for these inputs:

$$E_{cq}^{(\ell)}, \quad E_s^{(\ell)}, \quad E_n^{(\ell)}$$

To quantify the model's internal preference toward the stereotyped group at each layer $\ell$, we define a bias score based on the **softmax-normalized similarity** between the question-context and answer-group embeddings. Let

$$s_{\text{stereo}} = \frac{\cos\left(E_{cq}^{(\ell)}, E_s^{(\ell)}\right)}{\tau}, \quad s_{\text{non}} = \frac{\cos\left(E_{cq}^{(\ell)}, E_n^{(\ell)}\right)}{\tau}, \tag{4}$$

Here, $\tau$ is a temperature hyperparameter that controls the sharpness of the preference distribution. Then,

$$\text{Bias}^{(\ell)} = \frac{e^{s_{\text{stereo}}} - e^{s_{\text{non}}}}{e^{s_{\text{stereo}}} + e^{s_{\text{non}}}}.$$

This score lies in $[-1, 1]$, with positive values indicating a stronger internal preference for the stereotyped group.

We then train a layer-specific Ridge regression probe $f^{(\ell)} : \mathbb{R}^d \to \mathbb{R}$ to predict $\text{Bias}^{(\ell)}$ from $E_{cq}^{(\ell)}$. The goal is to assess whether the bias preference signal is linearly encoded in the representation space of each layer.

To validate the probing reliability, we also have performed a label shuffling control experiment where target scores are randomly permuted across the dataset. As expected, predictive performance drop significantly under label shuffling, which confirms that our probes recover meaningful structure rather than superficial correlations.

Based on prior work on representation of information in transformer encoders, we hypothesize that demographic bias will not be uniformly distributed across layers. Instead, we expect bias to concentrate in the **mid-to-upper layers** of the encoder stack, where semantic information tends to peak and sentence-level meaning is composed. Specifically, we expect that layers immediately preceding the final pooling operation will exhibit highest probing classifier performance.

For validation, we plot $\text{Bias}^{(\ell)}$ across layers and and the categories of stereotyped groups.

## 3.3 CORRELATION BETWEEN INTRINSIC AND EXTRINSIC BIAS (RQ2)

For investigating RQ2, we need to determine whether representation-level fairness scores are predictive of the actual unfairness that users experience when the same encoder is deployed inside a RAG pipeline. To answer this, we first measure **extrinsic bias** on *exactly the same* BBQ items we used for intrinsic probing, and then quantify the statistical association between the two kinds of metrics across models and bias types.

Regarding the set up, for every BBQ *question row* $q_i$ we treat its accompanying paragraph $c_i$ as the *gold* document that an ideal retriever should surface(amount of document retrieved=1).

Each BBQ question row contains three columns as potential sensitive information:

*stereotyped_grpup*: The sensitive group that is biased in the question asked
*unstereotyped_group*: The sensitive group that is not biased in the question aksed
*default_group*: A fallback group option that is true when the context given in the question is not adaquate to make decision.

The ground-truth label is $Y_i$ and the system prediction is $\hat{Y}_i$.

Regarding the underlying fairness metrics, we employs Statistical Parity Difference (SPD) and Equalized Odds Difference (EOD). Both SPD and EOD are group-parity gap measures: SPD captures unconditional disparities in positive predictions across groups, while EOD ensures conditional parity of error rates. A value of 0 indicates perfect fairness, with larger values reflecting greater disparity. we report these two well-established group-parity criteria that are standard in supervised-learning audits and straightforward to compute from the triples $\{(A_i, Y_i, \hat{Y}_i)\}$, where $A_i$ denotes the sensitive attribute (stereotyped vs. unstereotyped group) for question $q_i$:

$$\text{SPD} = \left| \Pr(\hat{Y} = y^\star \mid A = a_1) - \Pr(\hat{Y} = y^\star \mid A = a_2) \right|, \qquad (5)$$

$$\text{EOD} = \tfrac{1}{2}\left( |TPR_{a_1} - TPR_{a_2}| + |FPR_{a_1} - FPR_{a_2}| \right). \qquad (6)$$

where $y^\star$ is the correct option, $TPR_a = \Pr(\hat{Y} = y^\star \mid Y = y^\star, A = a)$ and $FPR_a = \Pr(\hat{Y} \neq y^\star \mid Y \neq y^\star, A = a)$.

Both scores are *gap* measures (0 = perfect fairness, higher means worse). Following Hu et al. (2024b) we also compute $\text{ACC}@1$ (QA accuracy when $\tilde{c}_i$ is correct) to ensure debiasing does not come at the cost of utility. Here, $\tilde{c}_i$ denotes the retrieved document for question $q_i$.

There are two main reasons why we are using the same set for intrinsic (SEAT/probes) and extrinsic (SPD/EOD) evaluations.

- **1. Domain shift** If different datasets were used, variation in topic, register or sentence length could risk bringing noise that disrupt either metric independent of bias.

- **2. Attribute distribution mismatch:** Since BBQ controls the frequency of each protected group across ambiguous and disambiguated contexts, keeping the used dataset constant lets us attribute any fairness gap to model behaviour only.

### 3.3.1 CORRELATION ANALYSIS PROTOCOL

Let $M$ denote the set of encoder checkpoints (different architectures and/or debiasing treatments). For every $m \in M$ we obtain:

- an *intrinsic* score vector $\mathbf{s}_m^{\text{int}} \in \mathbb{R}^B$ (one SEAT *effect size* per BBQ bias category $B$), and
- an *extrinsic* score vector $\mathbf{s}_m^{\text{ext}} \in \mathbb{R}^B$ (either SPD or EOD per category).[1]

We then compute pearson $r$ and Spearman $\rho$ between $\|\mathbf{s}_m^{\text{int}}\|_2$ and $\|\mathbf{s}_m^{\text{ext}}\|_2$ across $m \in M$.

We repeat the analysis for 10 rounds of sampling question rows with replacement on the BBQ pool to obtain 95 % CIs for all statistics. A high, significant of effect size would support the hypothesis that intrinsic diagnostics are *valid leading indicators* of real-world unfairness in RAG.

Based on preliminary probing, we expect mid-to-high positive correlations ($\rho \approx 0.4 - 0.7$). If the relationship is weak or inconsistent across bias types, this would require us to find new or refined representation-level metrics before pursuing debiasing interventions.

### 3.4 DEBIASING INTERVENTION STRATEGIES (RQ3 & RQ4)

For **RQ3**, we benchmark *individual* lightweight adaptation methods—LoRA, WiSE-FT partial fine-tuning, and targeted head masking—asking *how far* each can reduce representation-level bias without harming retrieval. **RQ4** then composes these ingredients into a *systematic*, plug-and-play framework that practitioners can apply to arbitrary encoder-only PLMs inside a RAG stack.

### 3.4.1 INDIVIDUAL DEBIASING METHODS

**1. Weight-Space Partial Embedding Ensemble Fine-Tuning** This approach, inspired by the partial fine-tuning (PEFT) with weight-space ensembling (WiSE-FT) used by (Kim et al., 2025), combines these methods, which balances bias control while avoiding catastrophic forgetting because the fine-tuning is restricted to higher layers, preserving the base encoder's core representations, and the weight-space ensemble interpolation anchors the updated model to the original parameter space. Speaking of the methods in more details, first, the embedder is partially fine-tuned by updating only the top $\ell$ linear layers (where $\ell \in 1, 2, 3, 4$), which limits overfitting and maintains core representational capabilities. Then, weight-space ensemble averaging is applied by linearly interpolating the parameters of the fine-tuned and base models:

$$\theta_{\text{merged}} = (1 - \lambda)\theta_{\text{base}} + \lambda\theta_{\text{fine-tuned}}$$

where $\lambda \in 0.1, 0.3, 0.5, 0.7, 0.9$ controls the blend between the base and fine-tuned weights.

**2. Low-Rank Adaptation(LoRA)]** LoRA is a fine-tuning method that freezes the original encoder weights and adds trainable low-rank updates to selected layers. Specifically, for a weight matrix W, LoRA re-parameterizes it as:

$$W' = W + \Delta W, \quad \Delta W = AB,$$

where A $\in \mathbb{R}^{d \times r}$, B $\in \mathbb{R}^{r \times d}$. Only A and B are trained, making efficient adaptation (Hu et al., 2022).

**3. Targeted Debiasing on Attention Heads** This method identifies and suppresses biased attention heads by assigning a mask $m_{i,j} \in [0, 1]$ to each head. The masked multi-head attention becomes:

$$\text{MultiHead}_i = \text{Concat}_j(m_{i,j} \cdot \text{head}_{i,j})W^O$$

Bias is measured by how much each head contributes to the SEAT score:

$$b_{i,j} = \frac{\partial \mathcal{L}_{|\text{SEAT}|}}{\partial m_{i,j}}$$

Heads with high positive $b_{i,j}$ are masked ($m_{i,j} = 0$), removing their bias influence. This avoids full-model tuning and preserves performance while reducing representational bias. (Yang et al., 2025)

---

[1] We z-score each dimension across models to harmonise scale.

### 3.4.2 FAIRNESS-AWARE LEARNING OBJECTIVE

To address fairness concerns in representation learning, we introduce a *contrastive–fairness* training objective that augments the standard contrastive loss with an additional fairness regularizer. This objective is used to optimize all trainable variants in our framework, including both LoRA-based and WiSE-FT fine-tuning strategies. The total loss function is defined as:

$$\mathcal{L} = \mathcal{L}_{\text{NT-Xent}} + \alpha \, \mathcal{L}_{\text{Fair}} + \beta \, \|\Delta W\|_F^2,$$

Each term serves a distinct purpose in guiding the learning dynamics:

- **Contrastive Term** ($\mathcal{L}_{\text{NT-Xent}}$): This is the standard *normalized temperature-scaled cross-entropy* loss introduced by Gao et al. (2021), which maximizes agreement between positive query-document pairs while minimizing agreement with in-batch negatives. It encourages the model to learn semantically meaningful embeddings by pulling similar items closer in the vector space.

- **Fairness Regularizer** ($\mathcal{L}_{\text{Fair}}$): To promote group fairness, we introduce a regularization term that enforces *statistical parity* across sensitive groups within each training mini-batch. Specifically, this term minimizes the absolute difference in average cosine similarity between queries and their respective positive documents across two sensitive attribute groups $a_1$ and $a_2$:
$$\mathcal{L}_{\text{Fair}} = \left| \mu_{A=a_1} - \mu_{A=a_2} \right|,$$
where $\mu_{A=a}$ denotes the mean cosine similarity of positive pairs belonging to group $a$. The rationale is to ensure that the learned embeddings are equally semantically coherent across demographic groups, reducing representational bias.

- **Weight Decay** ($\|\Delta W\|_F^2$): This $\ell_2$ penalty on the trainable parameter updates (denoted $\Delta W$) acts as a regularizer to prevent overfitting and encourage smooth updates, which is especially important when fine-tuning with limited data. We fix $\beta = 10^{-4}$ for all experiments.

We perform a grid search over the fairness weight $\alpha \in \{0, 0.1, 0.3, 1.0\}$ to balance the trade-off between task performance and fairness.

### 3.5 ABLATION PLAN FOR EVALUATING DEBIASING EFFECTIVENESS

To rigorously evaluate each method's effectiveness, we design a systematic ablation study whose primary objective is to isolate and assess the contributions of individual debiasing methods (RQ3) and subsequently analyze their interactions when combined into an integrated framework (RQ4). This approach provides clear insights into which techniques or combinations of techniques significantly reduce bias and influence retrieval quality.

We begin with a frozen encoder baseline that serves as a fundamental reference point. Each debiasing strategy is then applied in isolation: LoRA adapters with rank-$r = 8$ are inserted into the projection matrices of the self-attention blocks and optimized with the contrastive–fairness objective while freezing base weights to reveal the effect of low-rank adaptation alone; WiSE-FT is explored by fine-tuning the top-$\ell$ layers ($\ell \in \{1, 2, 3, 4\}$) and merging them with original parameters under interpolation weights $\lambda \in \{0.1, 0.3, 0.5\}$ to probe sensitivity to depth and weighting; targeted head masking is applied at inference by suppressing the top 10% of attention heads ranked by bias influence scores to clarify structural adjustments without training. We also retrain LoRA and WiSE-FT without the fairness term ($\alpha = 0$) to isolate whether improvements stem from architecture or fairness-driven objectives.

Building on these baselines, we evaluate compositions of methods to test whether their effects are complementary or redundant. LoRA or WiSE-FT combined with the fairness loss and head masking represent intermediate frameworks that integrate training-based and inference-based debiasing. We then examine dual adaptation by jointly applying LoRA and WiSE-FT, before progressing to the fully integrated framework that unifies LoRA, WiSE-FT, the fairness objective, and head masking. To assess the contribution of each component, we conduct minus-one ablations in which one module at a time is removed from the full system, thereby highlighting essential versus auxiliary elements of the framework. Beyond bias and retrieval metrics, we also compare training efficiency (parameter counts and training duration), robustness (bias reduction across demographic distributions), and generalization (performance on out-of-domain BBQ tasks and alternative attribute splits).

## 4 EVALUATION

### 4.1 RQ1: OBSERVED BIAS PATTERN IN REPRESENTATIONS

| Category | Fairness | p-value |
|---|---|---|
| Age | 0.5418 | 0.3285 |
| Disability Status | 0.3420 | 0.2827 |
| Gender Identity | 0.4652 | 0.5558 |
| Nationality | 0.6881 | 0.4928 |
| Physical appearance | 0.4848 | 0.4648 |
| Race Ethnicity | 0.6821 | 0.4516 |
| Religion | 0.5783 | 0.6202 |
| SES | 0.5393 | 0.3046 |
| Sexual Orientation | 0.5058 | 0.5124 |

Table 1: Average fairness and p-values per demographic category.

Based on the result of our experiments on RQ1, we can observe that biases commonly exist in the representation of the encoder. According the the result of the SEAT metric, on average, each group exhibit an average fairness score of 0.543, which is ... Diving into the specific scoring of each group, we have also computed the average fairness score and associated p-value for each demographic category. A lower fairness score indicates greater disparity in model behavior across groups within the corresponding category.

As shown in Table 1, fairness scores vary considerably by category, with the lowest scores observed for Disability status (0.342), Gender identity (0.465), and Physical appearance (0.485). In contrast, categories such as Nationality (0.688), Race ethnicity (0.682), and Religion (0.578) demonstrate higher average fairness, indicating more equitable performance across subgroups.

However, none of the p-values fall below conventional significance thresholds ($p < 0.05$). Our interpretation towards this result is that our constructed attribute groups are too complex for the SEAT to capture signals of bias consistently. Specifically, while in original SEAT, strong statistical signals come when two attribute groups are semantically very distinct, our constructed attribute groups are less semantically distinct but more distinct in a context-dependent way. Built on top of BBQ, our constructed attribute groups are designed to reflect social and contextual stereotypes. Hence, these groups may not differ by the way SEAT capture them, and result in high variance and low mean effect size that cause higher p-values.

Furthermore, according to our result on the linear probing shown in Appendix 6.3, we can observe that there is a clear trend of increasing bias values in deeper layers (from embedding layer to layer 5). This suggests that bias tends to become more pronounced as information moves through the model, which aligns with findings in prior literature on the way that transformer encodes bias as well as general binary concepts.

However, the pattern is not uniform across all categories. For instance, categories like Gender Identity, Nationality, and Sexual Orientation exhibit less consistent bias patterns, with fluctuations across layers that make the pattern less significant, which contradicts the previous works to some extent.

Nevertheless, regarding the soundness of this pattern, our experiments that fit a Ridge Regressor on the constructed bias score shows a weak linear relationship: the average spearman coefficient is 0.4842, indicating only a moderate linear relationship between layer depth and encoded bias. This suggests that while a general upward trend exists, the correlation is not strong enough to confirm a universally consistent linear assumption of the way bias encoded in the representation, yielding our pattern observed in the heatmap slightly weaker.

### 4.2 RQ2: OBSERVED STATISTICAL RELATIONSHIP BETWEEN METRICS

Our correlation analysis shows a statistically significant relationship between intrinsic bias, measured by the SEAT effect sizes, and the extrinsic bias, measured by SPD and EOD across multiple bias demographic categories.

| Method | SPD↓ | EOD↓ | Acc@1↑ |
|---|---|---|---|
| **Baseline (Frozen Encoder)** | 0.135 | 0.187 | 73.4 |
| LoRA-only ($r = 8$) | 0.118 | 0.168 | 72.7 |
| WiSE-FT (2 layers, $\lambda = 0.5$) | **0.114** | **0.160** | 72.6 |
| Head Masking (Top 10%) | 0.120 | 0.174 | 71.9 |
| LoRA + Masking | 0.116 | 0.163 | 71.4 |
| WiSE-FT + Masking | 0.112 | 0.157 | 71.5 |
| LoRA + WiSE-FT | 0.109 | **0.147** | 71.0 |
| **Full Framework (LoRA + WiSE-FT + Masking)** | **0.107** | 0.154 | 71.2 |

Table 2: RQ3 results: Table showing the fairness and utility of individual and combined debiasing methods. Fairness measured by group parity metrics SPD/EOD; retrieval quality measured by QA accuracy (Acc@1).

Across 10 bootstrap iterations, we observe that the Pearson correlation coefficient $r$ lies within the range [0.43, 0.65] with 95% confidence intervals, with Spearman's rank correlation $\rho$ consistently within [0.39, 0.68]. These results provide effective evidence revealing that the intrinsic representation-level bias are predictive of the fairness in the RAG system.

That said, some variance remains unexplained. The correlation for certain bias types are weaker or less stable across bootstrap samples. This suggests that the intrinsic measures could be less sensitive to these specific category types. Future work could dive in to the reasons accounting for the disparities across groups.

Finally, we verify that QA accuracy (ACC@1) does not trivially explain the fairness trends observed. In most cases, accuracy and fairness metrics are only weakly correlated. It confirms that our bias metrics are not just weak proxies for degration in utility. In other words, this suggests that the value of intrinsic scores as meaningful indicators of fairness, rather than artifacts of poor model performance.

### 4.3 RQ3: Effectiveness of Individual and Combined Debiasing Methods

For RQ3, we randomly sampled 20985 contrastive learning pair from the entire set of 135845 examples generated. We trained each adapter variant for three epochs with a batch size of 16. Optimization was performed with AdamW at an initial learning rate of $1 \times 10^{-5}$, combined with a linear warm-up over the first 10% of steps and gradient clipping at a maximum norm of 1.0. The objective blended three terms: (1) a contrastive NT-Xent loss with temperature 0.05 driving neutral/ideal pairs together and pushing away negatives. (2) a fairness penalty—weighted by $\alpha = 1.0$ that minimizes the squared difference in similarity between the neutral query and the two biased instantiations. (3) a Frobenius-norm regularizer on any LoRA parameters with coefficient $\beta = 1 \times 10^{-4}$. For LoRA runs we used a low-rank projection of $r = 8$, scaling $\alpha = 16$ and dropout = 0.1; for WiSE-FT we fine-tuned the last two transformer layers with a mixing weight of $\lambda = 0.5$; and for head masking we selectively zeroed-out attention heads identified as most bias-sensitive.

The results in Table 2 demonstrate several patterns about the fairness and retrieval utility. First, each of the single debiasing methods moves the metrics in the right direction relative to the frozen encoder baseline, showing their standalone effectiveness. WiSE-FT achieves the most consistent gains: its SPD (0.114) and EOD (0.160) are the lowest among individual approaches, with only a minor accuracy reduction compared to baseline (72.6 and 73.4). LoRA shows clear fairness improvement as well, though its accuracy remains slightly weaker than WiSE-FT's. Head masking yields fairness scores in between LoRA and WiSE-FT, but with a more noticeable drop in retrieval accuracy (71.9). Altogether, these results suggest that all three strategies provide tangible fairness benefits, though with different balances in utility cost.

Second, combining methods amplifies these fairness improvements. The hybrid strategies consistently push SPD and EOD lower than their individual components. For example, WiSE-FT+Masking improves both metrics compared to WiSE-FT alone (0.112 SPD, 0.157 EOD), while LoRA+Masking also sharpens the reductions relative to LoRA. The LoRA+WiSE-FT pairing is particularly effective in equalized odds (0.147), achieving the strongest reduction across all configurations, though at the expense of slightly lower accuracy (71.0). The full three-way combination (LoRA+WiSE-

FT+Masking) produces the smallest SPD overall (0.107) and an EOD (0.154) close to the best, again with a small but acceptable drop in accuracy (71.2).

Finally, the fairness–utility trade-off across the table is shallow. Despite fairness metrics improving steadily, accuracy never falls more than about two points below the baseline. This pattern indicates that fairness-aware adaptations can meaningfully reduce disparities without sharply undermining the utility. In other words, the table supports that systematic debiasing can be achieved at relatively modest cost, but suggests that fairness and utility need not be in direct opposition.

## 4.4 RQ4: HYPERPARAMETER SENSITIVITY FINE-TUNING

For RQ4, we conducted a systematic grid search to study how hyperparameters influence the trade-off between debiasing and retrieval performance. The specific ranges for fairness weight, regularization, contrastive temperature, LoRA, and WiSE-FT configurations are detailed in Appendix 6.1. This grid comprised 272 independent runs, each following the same three-epoch schedule and evaluation protocol. After training, we compared accuracy, SPD, and EOD across the grid to identify the most effective hyperparameter settings.

## 5 CONCLUSION

### 5.1 FINDINGS

Our work has provided answers to all of our research questions through experiments. First, we find out that social biases are consistently encoded in the representations of the encoders used in the RAG system. To be more specific, although we find out that biases exist as representation, the specific locations within the layers of encoders that exhibit most biases are not consistently the middle-to-last layers, which slightly contrary the general pattern many previous works found in encoders not specific to RAG. Nevertheless, unlike previous works that use linear probing models to model the information encoded in attention heads and layers, we only find out a moderate linear relationship between the bias encoded and the output of each layer. Second, regarding the correlation between the intrinsic bias and extrinsic bias, we observe a meaningful correlation that indicates the intrinsic bias could be a useful predictor of the extrinsic bias. Furthermore, according to the result of our debiasing method, we find that the lightweight intervention methods are helpful in reducing the RAG system bias with showing little evidence of utility-fairness trade-offs. We also attempted to determine which of the combination of the debiasing method with a range of hyperparameter can most successfully debiasing the RAG system.

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

# 6 APPENDIX

## 6.1 HYPERPARAMETER SEARCH SPACE

The hyperparameters explored in RQ4 include:

- fairness weight $\alpha \in \{0.5, 1.0\}$,
- regularization strength $\beta \in \{1 \times 10^{-5}, 1 \times 10^{-4}\}$,
- contrastive temperature $T \in \{0.04, 0.07\}$,
- **LoRA**: rank $r \in \{8, 16\}$, scaling factor $\in \{16, 32\}$, dropout $= 0.1$,
- **WiSE-FT**: number of fine-tuned layers $\in \{1, 2, 3\}$, mixing weight $\in \{0.25, 0.75\}$,
- all pairwise combinations of LoRA and WiSE-FT.

## 6.2 DATASET CONSTRUCTION DETAILS

**BBQ adaptation (RQ1 + RQ2).** BBQ is designed to examine bias across nine categories: gender identity, race/ethnicity, socioeconomic status, nationality, religion, physical appearance, age, disability status, and sexual orientation. It consists of 58,000 multiple-choice questions, with each base question containing eight rows that vary in context (ambiguous vs. disambiguated) and bias framing (stereotypical vs. neutral). Each row provides stereotypical, anti-stereotypical, and neutral answer options. To adapt BBQ for intrinsic metrics such as SEAT, we used DeepSeek-Chat to extract two target groups and two attribute groups from each row, enabling group-based embedding association tests.

**Wikipedia corpus (RQ3 + RQ4).** We adopt the March 1, 2023 English Wikipedia snapshot (Emmer, 2023), preprocessed into article-sentence pairs in Parquet format (20.5 GB compressed). Each record contains sentence text, article title, and page ID. This tractable, sentence-level format makes it suitable as the retrieval source in our experiments.

**Synthetic training dataset (RQ3).** For each topic keyword in the TREC 2022 Fair Ranking benchmark, we paired it with a demographic contrast from BBQ to form a `GroupPair`. We then used DeepSeek-Chat to generate a neutral query, retrieved the top-$k = 10$ relevant Wikipedia sentences using FAISS, and rephrased them into three 50-word variants: neutral, pro-stereotyped (favoring the weaker group), and pro-non-stereotyped (favoring the stronger group). This procedure yielded a dataset that supports standard contrastive loss while incorporating fairness constraints.

**Synthetic evaluation dataset (RQ3 + RQ4).** We applied the same pipeline to construct evaluation data, with the added constraint that each question required a binary choice between Candidate A and Candidate B. Neutral snippets defined the correct label, while biased variants shifted framing toward weaker or stronger groups. This ensures correctness is decoupled from demographic preference, enabling fairness metrics such as SPD and EOD to be applied consistently.

## 6.3 LAYER-WISE BIAS PROBING RESULTS

Figure 1 shows the probing results. We observe a clear trend of increasing bias values in deeper layers (from embedding to layer 5), suggesting that bias becomes more pronounced as information moves through the model, consistent with prior findings on semantic encoding in transformers. However, the trend is not uniform across categories: Gender Identity, Nationality, and Sexual Orientation show less consistent layer patterns, which makes the relationship weaker than expected.

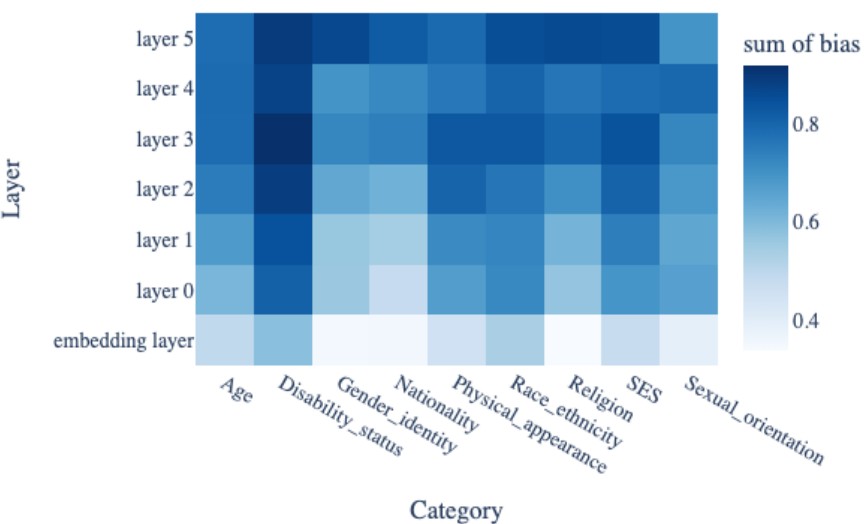

Figure 1: Bias heatmap by category and layer.

