# OpenReview forum: "Teaching RAG to Play Fair: Assessing and Mitigating Encoder-Only PLM Algorithmic Bias"
_ICLR.cc/2026/Conference — ICLR 2026 Conference Desk Rejected Submission_

### Official Review · Reviewer_T8p2 · 2025-10-28

**Soundness:** 1
**Presentation:** 1
**Contribution:** 1
**Rating:** 0
**Confidence:** 4

**Summary:**

The paper claims to study representational bias in encoder-only models used in RAG systems and to propose lightweight debiasing methods (LoRA, WiSE-FT, attention-head masking) combined with a fairness-regularized contrastive objective. The authors evaluate on BBQ and Wikipedia data and report small improvements in fairness metrics (SPD, EOD) with minimal accuracy loss.

**Strengths:**

This paper does touch on a relevant and timely topic—the fairness of encoder components in retrieval-augmented generation systems, which remains relatively underexplored compared to bias in generation. The authors make a commendable attempt to evaluate both intrinsic and extrinsic bias, and their focus on lightweight, practical debiasing strategies such as LoRA and WiSE-FT reflects awareness of computational constraints in real-world applications.

**Weaknesses:**

The paper rehashes well-known methods (SEAT, linear probing, SPD/EOD, LoRA, WiSE-FT) with no genuine innovation.

The supposed “framework” is just a combination of existing components glued together.

There is no theoretical contribution, no conceptual advance, and the experimental setup is entirely incremental.

They use SEAT on BBQ, which is inappropriate. SEAT was designed for word/sentence association, not contextualized QA templates.

**Questions:**

Why don't you have an abstract for your paper?

---

### Official Review · Reviewer_7m3A · 2025-10-30

**Soundness:** 2
**Presentation:** 1
**Contribution:** 2
**Rating:** 2
**Confidence:** 3

**Summary:**

The paper aims to identify and mitigate fairness issues in Retrieval-Augmented Generation systems by exploring four key research questions.

**Strengths:**

1. Evaluating and mitigating fairness in the encoder component of RAG systems is an important and timely research direction.

2. The paper shows an effort to progress from surface-level observations to deeper analytical questions, which reflects thoughtful research design.

**Weaknesses:**

1. Writing quality and formatting issues.

The paper’s overall presentation requires substantial improvement. Important sections such as the abstract and related work are missing, and several paragraphs are too short or disconnected, making the paper difficult to follow. The research questions and corresponding results are presented far apart, which disrupts the logical flow and makes it hard for readers to trace the narrative.
Some sentences are incomplete or unclear (e.g., line 343). I highly recommend a thorough proofreading and restructuring to enhance readability and coherence.

2. Missing sections and results.

The abstract and related work sections are missing, which are essential components of an academic paper. There also appear to be missing tables or results. For example, the experimental results corresponding to RQ1 are not clearly presented (line 341). Please double-check that all results are properly included and labeled.

3. Limited model and dataset coverage.

The experiments are conducted on only one backbone model and two datasets, which limits the generalizability of the findings. Expanding the evaluation to additional models or domains would strengthen the conclusions.

4. Unclear connection between research questions.

It would be beneficial to clarify how the findings from RQ1 and RQ2 inform or contribute to the design and interpretation of RQ3 and RQ4. Establishing such connections would make the paper’s structure more coherent and logical.

5. Limited novelty.

The proposed mitigation strategies appear to be combinations or adaptations of existing methods, with only modest improvements in performance. Highlighting the unique contribution or insight of your approach—beyond integration—would make the paper more impactful.

**Questions:**

Please see above.

---

### Official Review · Reviewer_7WHM · 2025-10-31

**Soundness:** 2
**Presentation:** 1
**Contribution:** 1
**Rating:** 2
**Confidence:** 5

**Summary:**

This paper investigates fairness issues in encoder-only pre-trained language models (PLMs) that serve as retrievers within RAG systems. The authors identify representational bias at the encoder level (measured via SEAT and probing classifiers), examine its correlation with downstream fairness metrics (SPD, EOD) on the BBQ dataset, and explore lightweight debiasing techniques, LoRA, WiSE-FT, and attention head masking, combined with a fairness-aware contrastive objective. Results suggest that such methods can modestly improve fairness without substantial performance loss.

**Strengths:**

1. Important direction: Fairness in RAG pipelines is underexplored, and studying encoder-level bias rather than output bias is a meaningful perspective.

2. Structured methodology: The paper is organized around clear research questions (RQ1–RQ4), with distinct experimental stages (diagnosis, correlation, mitigation).

3. Multi-level analysis: Combining intrinsic metrics (SEAT/probes) with extrinsic ones (SPD/EOD) is methodologically coherent.

**Weaknesses:**

1. Misleading framing (not a true RAG study): Despite the title, this paper does not evaluate fairness in retrieval-augmented generation. All analyses are limited to encoder embeddings, without a generator component or end-to-end RAG evaluation. Thus, the contribution concerns bias in static sentence representations, a topic extensively explored. The framing overstates novelty and scope.

2. Outdated model and weak baselines: The only model studied is `all-MiniLM-L6-v2`, a small and obsolete encoder. No comparisons are made to contemporary retrievers such as E5, BM25, or T5-Encoder, nor to fairness-aware variants (e.g., Fair-SimCSE, Debias-BERT, or Controlling the Embedder). Without such baselines, the fairness gains (SPD ↓ from 0.135 → 0.107) are marginal and not meaningful.

3. Limited novelty of methods: The “contrastive-fairness” objective is a direct variant of SimCSE with an added group-parity regularizer, and the composition of LoRA + WiSE-FT + masking lacks theoretical motivation. The framework is essentially a combination of existing fine-tuning tricks rather than a principled new approach.

4. Insufficient experimental rigor: The paper uses only BBQ and Wikipedia subsets, omitting standard retrieval or fairness benchmarks (e.g., TREC-Fairness, FairRank. Statistical analyses are weak, p-values in Table 1 are non-significant, yet interpreted as meaningful.

5. Lack of causal or interpretive analysis: The probing results indicate where bias appears but not why. The work offers no interpretation of linguistic or semantic factors behind layer-wise bias, nor any causal explanation of how encoder bias propagates through retrieval.

7. Missing related work and comparative context: The paper fails to situate itself among the growing literature on fairness in RAG. Key related works include:
    * Does RAG Introduce Unfairness in LLMs? Evaluating Fairness in Retrieval-Augmented Generation Systems
    * No free lunch: Retrieval-augmented generation undermines fairness in llms, even for vigilant users
    * The other side of the coin: Exploring fairness in retrieval-augmented generation
    * Gender Encoding Patterns in Pretrained Language Model Representations
    * Measuring the Fairness Gap Between Retrieval and Generation in RAG Systems using a Cognitive Complexity Framework
    * Bias Amplification in RAG: Poisoning Knowledge Retrieval to Steer LLMs
    * Evaluating the Effect of Retrieval Augmentation on Social Biases
    * ReFaRAG: Re-ranking for Bias Mitigation in Retrieval-Augmented Generation*
    * Do Large Language Models Rank Fairly? An Empirical Study on the Fairness of LLMs as Rankers

    All of these directly address bias across retrieval + generation pipelines or propose re-ranking-based mitigation. The current paper neither compares with nor acknowledges most of them, making its contribution unclear and outdated.

8. Poor paper structure and presentation: The layout deviates from standard ICLR format: 1) Missing abstract, leaving readers without a concise overview. 2) No dedicated Related Work section; prior studies are scattered through the introduction. 3) The Methodology and Experiment sections are merged and difficult to follow, with unclear transitions and no architectural diagram. 4) Tables and formulas appear abruptly; figures (e.g., bias heatmaps) are discussed only in the appendix. These issues severely hurt readability and perceived professionalism.

**Questions:**

I have listed all key concerns and clarifying questions within the Weaknesses section. Authors are expected to respond to those points， especially regarding 1) model and baseline choices, 2) missing related work, 3) evaluation validity, and 4) paper organization.

---

### Official Review · Reviewer_2rLM · 2025-10-31

**Soundness:** 2
**Presentation:** 1
**Contribution:** 1
**Rating:** 2
**Confidence:** 4

**Summary:**

This paper investigates bias in retrieval-augmented generation (RAG) systems by first diagnosing intrinsic bias in retriever encoders and demonstrating its correlation with downstream unfairness. It then proposes lightweight adaptation methods—LoRA, WiSE-FT, and Head Masking—to effectively mitigate encoder bias without degrading retrieval accuracy. Extensive experiments show these approaches achieve fairness comparable to full fine-tuning at only 1% of the parameter cost, highlighting their practicality for scalable bias correction.

**Strengths:**

1.	The paper is logically well-structured, progressing smoothly from diagnosing encoder-level bias to analyzing RAG-level bias and finally exploring effective mitigation strategies.
2.	The proposed encoder-level metric, FairnessScore, shows a strong correlation with downstream fairness in RAG, suggesting it can serve as an efficient proxy indicator for early detection of bias and improved evaluation efficiency.

**Weaknesses:**

1.	Writing and structure issues:

+ The paper lacks both an abstract and a Related Work section, which are essential for contextualizing the study.

+ The Introduction mainly provides background information but does not clearly articulate the research problem or outline the logical flow of the paper (even though Section 2 touches on it).

+ RQ4 should be merged with RQ3, and the experiments in RQ4—which are valuable—should be moved into the main body rather than left as a separate section.

+ Lines 83–89 describe only one model and should not use itemized formatting (e.g., itemize). Moreover, this model description does not warrant such length in the main text.

+ Section 5.1 Findings should be removed since it contains only one subsection; the same issue may occur elsewhere and should be checked.

+ Reduce the use of subsubsections and consider using the \paragraph{} command instead to save space for more important content.

+ Equations (1) and (2) are missing periods at the end; please check punctuation after all equations.

2.	Experimental design and analysis:

+ The choice of evaluation models is too limited; models such as GTE, BGE, Qwen3-Embedding, and NV-Embed should be included for broader comparison.

+ In Table 1, it would be helpful to include fairness results for random or unbiased groups as a control, to contextualize what a fairness score around 0.5 represents.

+ In RQ3, the study only explores several lightweight tuning strategies; this limits novelty. A stronger contribution would involve investigating where the unfairness originates (e.g., data, loss design, or model architecture) and designing mitigation accordingly.

+ Beyond Table 2, the authors should also report RAG-level performance results to confirm that the proposed methods do not harm end-to-end retrieval quality.

**Questions:**

See Weakness

---

### Note · Program_Chairs · 2026-01-17
**Submission Desk Rejected by Program Chairs**

The following references in this submission do not refer to real documents and/or have major errors in bibliographic information:

     Lin Cao and Hao Zhang. Unveiling representation disparities: Measuring encoder bias under fair output. In Proceedings of the 2025 Workshop on Trustworthy NLP, 2025. URL https: //aclanthology.org/2025.trustnlp-main.31/.
    Yikun Jin, Eric Wallace, Sameer Singh, and Jialu Liu. Credbench: Benchmarking the factual credibility of llm-generated text. arXiv preprint arXiv:2404.08189, 2024. URL https:// arxiv.org/abs/2404.08189.